# Transition Based Dependency Parser for Amharic Language Using Deep Learning

## Abstract

Researches shows that attempts done to apply existing dependency parser on morphological rich languages including Amharic shows a poor performance. In this study, a dependency parser for Amharic language is implemented using arc-eager transition system and LSTM network. The study introduced another way of building labeled dependency structure by using a separate network model to predict dependency relation. This helps the number of classes to decrease from 2n+2 into n, where n is the number of relationship types in the language and increases the number of examples for each class in the data set. Evaluation of the parser model results 91.54 and 81.4 unlabeled and labeled attachment score respectively. The major challenge in this study was the decrease of the accuracy of labeled attachment score. This is mainly due to the size and quality of the tree-bank available for Amharic language. Improving the tree-bank by increasing the size and by adding morphological information can make the performance of parser better.

## 1 Introduction

Amharic is one of the Ethiopian languages, grouped under Semitic branch of Afro-Asiatic language. Amharic serves as the official working language of the Federal Democratic Republic of Ethiopia and it is the second most spoken Semitic language in the world after Arabic with a total speakers of around 22 million, as per the census of 2007Asker (2010). Even though many works have been done to develop language-processing tools, they are mainly concentrated on English, European and East-Asian languagesNivre (2013).In spite of large number of speakers, Amharic is one of the under-resourced languages in that it needs many linguistic tools to be developed in general and dependency parser in particular. Researches like Maltparser attempted to develop universal dependency parser that can be used to parse sentences from different languages without making language specific modification to the systemE. (2007). Even though it is language independent dependency parser, few languages are incorporated in to Malt parser, such as English, Dutch, Turkish and ItalianE. (2007). Parsing in morphologically rich languages is challenging. This is because of the fact that, in morphologically rich languages, dependency relation exists between not only orthographic words (space-delimited tokens) but also relation within the word itselfNivre (2013). For example, in Amharic, an orthographic word combines some syntactic words into one compact string. These words can be function words such as prepositions, conjunctions and articles. This makes an orthographic word functioning as a phrase, clause or sentenceBinyam Ephrem Seyoum (2018). Studies on applying existing parser system on morphologically rich languages were conducted inAsker (2010) and marton; Nizar Habash; Owen Ranbow (2012). these studies showed applying parser like MST and malt parser on morphological rich languages results a poor performance. The first attempt to develop a dependency parser for Amharic language was made in Gasser (2010). Gasser used Extensible dependency grammar (XDG) to accomplish the work. However, their system was not tested on corpus data and that makes the system inapplicable. This aim of this study is to design and develop dependency parser for Amharic language that uses the rule of arc-eager transition system. The main contribution of this paper are: (i) designing a neural network model with LSTM and attention mechanism that gives a good accuracy, (ii) introducing a separate model for building labeled dependency structure, (iii) developing a dependency parser that can produce unlabeled and labeled dependency structure for Amharic sentences.

## 1.1 Dependency parsing

Dependency parsing is one of the techniques for analyzing syntactic structure of a sentence. It represents the head-dependent relationship between words in a sentence classified by functional categories or relationship labelsRangra (2015). The advantage of dependency parsing is that it provides a clear predicate argument structure, which is useful in many NLP applications that makes use of syntactic parsingNivre (2010). Approaches to perform dependency parsing are broadly grouped into grammar based and data driven approachesNivre (2010). In grammar-based approach, constructing dependency structure rely on explicitly defined formal grammar. In such approaches, parsing is defined as the analysis of a sentence with respect to the given grammarNivre (2010).Data driven approaches for dependency parsing uses a statistical method to build the dependency structure. In such approaches, the parser is trained on labeled dataset to learn the pattern of dependency relation, latter it uses its memory to parse new sentencesNivre (2010).Data driven dependency parser is further grouped in to transition based and graph based approaches. In transition-based dependency parsing, the parser uses transition systems to build a dependency structure. In this method, statistical model is trained on sequences of transition configurations or states for predicting transition action for new configurationNivre (2005). Then the parser uses the predicted sequence of transition actions as a guideline to build a dependency structure for a given sentenceNivre (2005).

## 1.2 Amharic language

Amharic is a Semitic language spoken in Ethiopia. It is the second wide mostly spoken Semitic language in the world next to Arabic Asker (2010). Amharic exhibits a root-pattern morphological phenomenon in which the root is combined with a particular prefix or suffix to create a single grammatical form or another stemDemeke & Getachew (2016)Asker (2010). Because of this, different authors use different number of tag sets to incorporate the created word forms. Some authors use a different tag sets for each of the morphological wordsDemeke & Getachew (2016) and other separates the clitic from the host word and assign a part of speech tag to host word and the clitic separatelyBinyam Ephrem Seyoum (2018). We used a treebank from provided inBinyam Ephrem Seyoum (2018), for training and constructing a network model. This treebank has 23 tag sets (see table 1) and uses the second method of assigning part of speech tag.

## 2 Methodology

The proposed Amharic dependency parser consists of two phases, which are training or learning phase and parsing phase, see figure 1. In the first phase, deep learning models are trained on Amharic treebank to learn the pattern of Amharic dependency relation. We divide this task in to two steps. The first step is training transition action predictor. This step is used to learn the pattern of arc-eager transition action on Amharic treebank. Once arc-eager transition predictor model is constructed, we also train another model for predicting dependency relationship type. The second phase is parsing phase in which the trained deep learning models are used for constructing a dependency structure to an Amharic sentence.

## 2.1 Learning relationship types

In this step, a network model, see figure 2, is trained on Amharic treebank to learn the pattern of arc-eager transition actions For doing this we convert Amharic treebank in to sequence of arc-eager transition configurations. A single configuration have five elements (stack state, stack part of speech tag state, buffer state, buffer part of speech tag state, and transition action), see figure 3. Each inputs then embedded to make a dense representation. That means words that are similar or used in similar grammatical places will have a closer numerical value. For example, the word 'werewwre' and 'gez' will have a closer weight than other words because they have similar grammatical place, in this case, they are used as a verb in the sentence. After each inputs are embedded, related inputs (stack of words state with stack of words POS state and buffer of words state with buffer of words POS state) are scalar multiplied using equation 1 and equation 2.

$P = W . T$ ………….. (1)
$Q = N . S$ ………….. (2)

Where $W$ is words in the stack, $T$ is the corresponding part of speech tags the stack, $N$ is words in

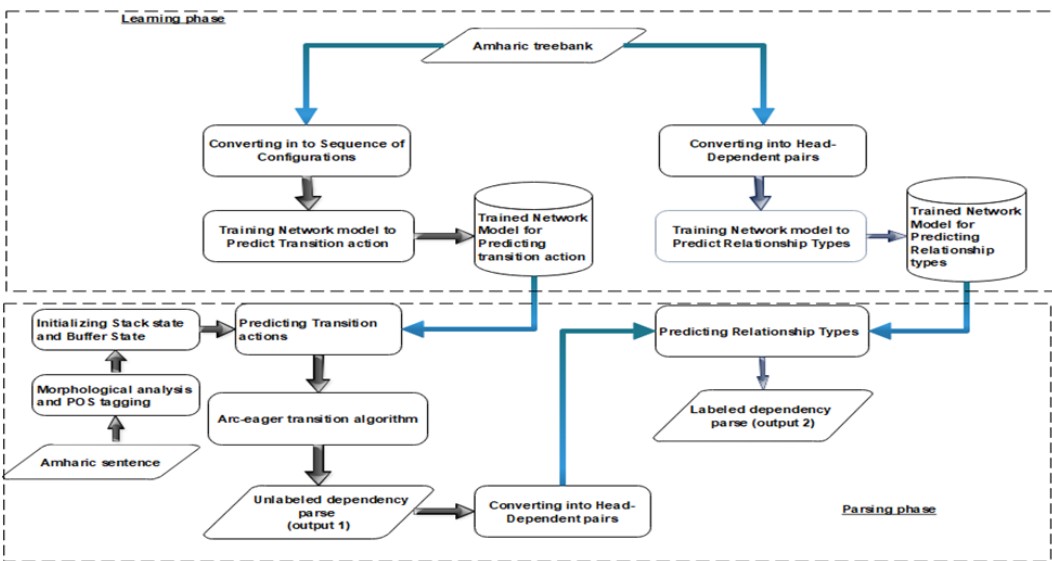

Figure 1: Architecture of the parser

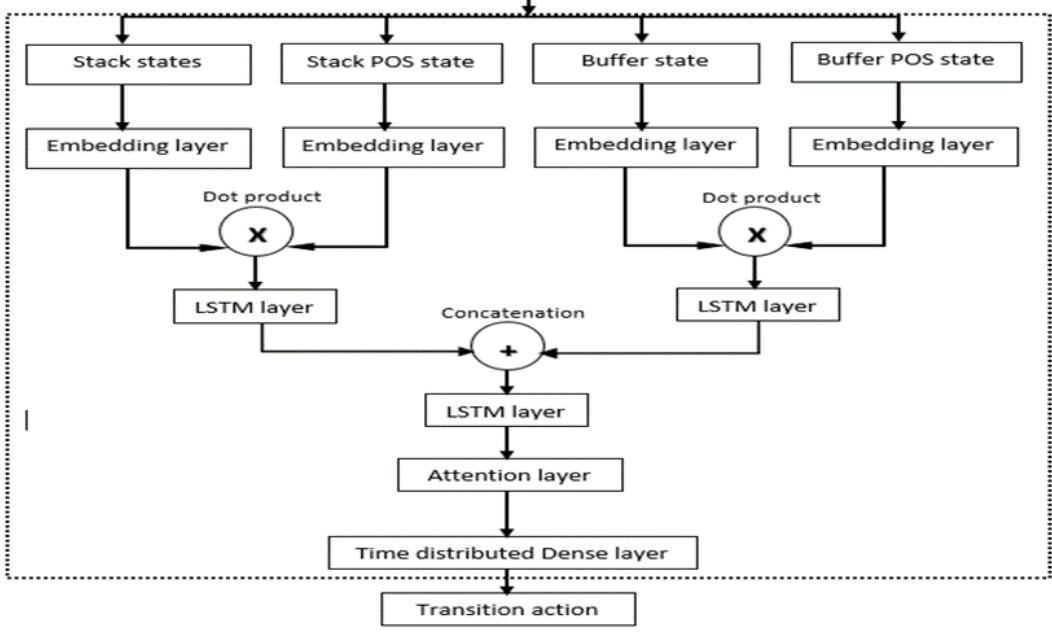

Figure 2: Network model of learning transition action

| Stack state | Stack POS state | Buffer state | Buffer POS state | Transition action |
|---|---|---|---|---|
| [-root-] | [-root-] | [ሰህን, አ, ን, ወረወር, አ ,ት, ።] | [NOUN, DET, ACC, VERB, SUBJC, OBJC, PUNCT] | Shift |
| [-root-] | [-root-] | [ፍየል, ጋሀ, አ, ል, አት, ።] | [NOUN, VERB, SUBJC, ADP, OBJC, PUNCT] | shift |

Figure 3: Sample instances of transition configuration

the buffer and $S$ is the corresponding part of speech tags in the buffer.

Before concatenating the stack state and buffer state, we used LSTM layer to make information to be connected from one-step to the next in the stack and buffer states.

The outputs of the two LSTM layers (LSTM layer for stack state and LSTM layer for buffer state) are concatenated. The concatenation layer simply combines matrix of stack state and buffer state. The last LSTM layer then applied to the concatenated stack state and buffer state, which are separately passes thought LSTM layer. This helps the information further connected from stake state to buffer state. Since arc-eager transition system keeps the word at stack before all its dependents at the buffer are seen, the transfer of information from stack state to buffer state helps the network to incorporate this phenomenon. The final layer of the network model is fully connected dense layer. It is applied to each time step of the outputs from the previous layer to decide the final output. The output of the network is one of the arc-eager transition actions (shift, left-arc, right-arc or reduce).

## 2.2 LEARNING RELATIONSHIP TYPE

Studies on transition based dependency parsing constructs unlabeled and labeled dependency parse from a given transition configurationE. (2007)Manning (2014)Dyer.That means they use 2n+2 classes during training the classifier, where 'n' is the total number of relationship types in the language. This method has a disadvantage that it increases the number of classes in the learning model so that the number of examples for each classes in the dataset will be small. This disadvantage can be even severing when the dataset is smaller. Besides that, as we observed in the experiment, the relationship type that holds between a dependent word and headword is not dependent on the transition configuration. That is because, transition configurations only give an information about which action to select among the four transition actions. On the other hand, the type of dependency relationship can be determined by using the part of speech tag of the dependent and headword. For example, the relationship type between '' and ' in the sentence '' '--- '' is det because of the part of speech tag of the word '' is DET.

In this study, we designed a separate model, as presented in figure 4, to learn the relationship type of head-dependent pairs in Amharic treebank. To do this we convert Amharic treebank into head-dependent pairs. The network model accepts part of speech tag of the dependent word and the part of speech tag of the corresponding headword. figure 5 shows sample of inputs to the network model For simplifying access to the network model, as shown in figure 5, ROOT (a word that is not exist in the original sentence) word is artificially added. This word is used to indicate the root word of a sentence during dependency construction.

Embedding layer is applied to the POS tag of dependent word and headword separately to represent it in a dense vector. The relationship between the dependent word '' and the headword '' is 'expl'. The relationship type 'expl' is used if the subject is explicitly indicated in the sentence, in our case, for instance the word ''. The other condition the subject is not explicitly indicated in the sentence. An example is given in figre 6 above. In this case, the relationship between the dependent word '' and the headword '' become 'nsubjc'. To handle this situation we applied LSTM layer. LSTM helps to transfer information from one time step to the next. In our case, it helps to pass information about the presence of subject word in the previous steps to select appropriate relationship type, either 'nsubj' or 'expl', for the head-dependent pairs accordingly. Finally, output is determined

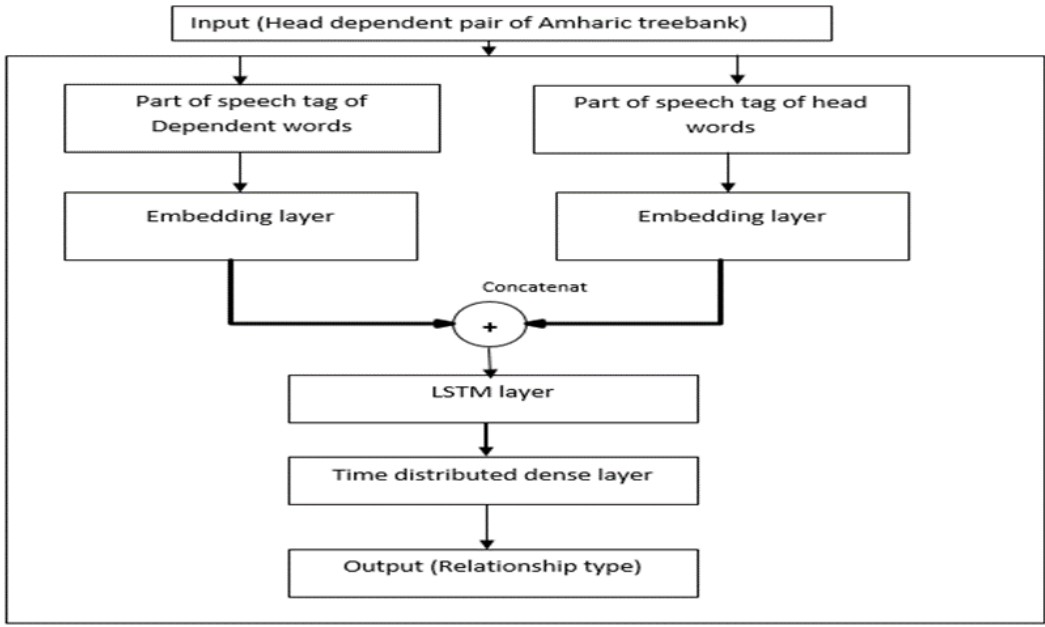

Figure 4: Network model for learning relationship type

| Dependent word | Dependent word's POS tag | Headword | Headword's POS tag | Class (relationship type) |
|---|---|---|---|---|
| እበበ | PROPN | በራ | VERB | nsubj |
| ቡሶ | NOUN | በራ | VERB | obj |
| በራ | VERB | ROOT | ROOT | root |
| ኸ | SUBJC | በራ | VERB | expl |

Figure 5: Sample inputs for the dependency relation predictor network model

| Dependent word | Dependent word's POS tag | Headword | Headword's POS tag | Class (relationship type) |
|---|---|---|---|---|
| ቡሶ | NOUN | በራ | VERB | obj |
| በራ | VERB | ROOT | ROOT | root |
| ኸ | SUBJC | በራ | VERB | nsubj |

Figure 6: Sample inputs for the dependency relation predictor network model

using time distributed dense layer. This layer outputs the predicted dependency label for a given head-dependent pairs of a sentence.

## 2.3 PARSING PHASE

The purpose of this phase is to construct a dependency relation for a given new sentence by the help of trained network models. the steps followed in parsing phase are shown in figure 1.

Step 1: Input. The input for the parsing system is an Amharic sentence. For example " "

Step2: Morphological analysis and part of speech (POS) tagging. In this step, morphological analysis and POS tagging is performed on the input sentence. We perform this task manually using the method presented on UD-Amharic treebank.

Step 3: Initializing stack state and buffer state. The network model for predicting transition action accepts four values (stack state, stack POS state, buffer state and buffer POS state). Due to this, the input sentence is used to initialize these states. The initial state of stack contains one element, 'root'. This element is used later as a headword for the root word of the sentence. The initial state of buffer contains all tokens of the input sentence. Stack POS state and buffer POS state contains corresponding POS tag of the words in the stack and the buffer respectively.

Step 4: Predicting transition actions. In this step, the network model trained for predicting transition action uses the value of stack state and buffer state to predict the one of the four arc-eager transition actions for the configuration. After the model select the transition action, the algorithm updates the state of the stack and the buffer according to the transition action and the rule of arc-eager transition system.

Step 5: Arc-eager transition algorithm. In this step, arc-eager transition rule is used to build unlabeled dependency relation for the given input sentences. The sequence of transition actions in the previous step are used to guide the algorithm to assert head-dependent relationship between the tokens and update the state of stack and buffer. When the buffer contains no element the algorithm stop the iteration and the constructed unlabeled dependency tree will be released.

Step 6: Converting into head-dependent pair. In this step, the unlabeled dependency parse is converted into head-dependent pairs to predict the dependency relation between the head-and dependent word.

Step 7: Predicting relationship type. In this step, the dependency relationship type of the head-dependent relation is predicted from the part of speech tag of head-dependent pairs using the network model trained for predicting dependency relationship type. The output of this step is labeled dependency structure of the given sentence.

## 3 EXPERIMENTAL RESULT

### 3.1 DATASET

In this study, we used the only Amharic treebank that is made available inBinyam Ephrem Seyoum (2018). It has 1074 sentences and we used additional 500 sentences that are manually constructed by the researcher with the help of linguistic experts at Jimma University.

### 3.2 EVALUATION OF TRANSITION ACTION PREDICTION

We experimented on 60/40, 70/30 and 80/20 train-test splitting ratios and achieved 93.1, 94.7 and 92.3 testing accuracy respectively for predicting transition action as shown in figure 8. Therefore, we select the 70/30 train-test splitting ratio throughout the experiments.

When we convert the treebank to transition configuration we got 26,242 configurations, among this we used 70 (with 1,102 sentences having 17,888 configurations) for training and 30 (with 472 sentences having 8,353 configurations) for testing. The dataset has 1475 unique words. We add other key words, such as –OOV- to indicate out of vocabulary words; -PAD- to indicate padding (zero) and 'root' to indicate the root word. Due to this, we got 1478 unique words. This makes the embedding layer for stack state and buffer state to have a dimension of 1478. On the other hand, the dataset has 23 unique part of speech tags. Here also, we added -OOV-, -PAD- and root key words and we have 26 unique key words for POS. This makes the embedding layer for POS state

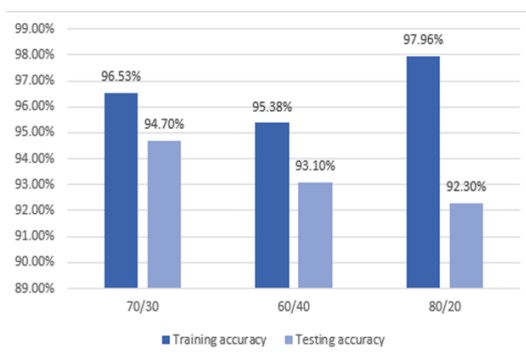

Figure 7: Accuracy of predicting transition action using different splitting ratio

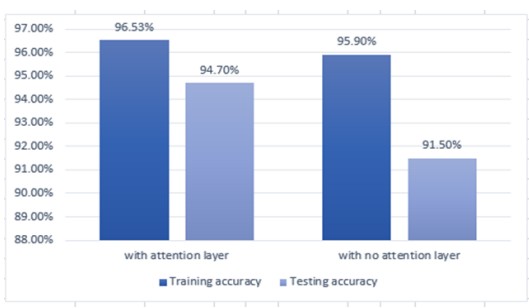

Figure 8: The accuracy of transition action prediction with and without adding of attention layer

of the stack and buffer to have 26 dimensions. For the LSTM layer we used 256 output dimensions and 128 dimensions for attention layer. For the output layer we used four output dimensions that is 0, 1, 2 and 3 (for Shit, Left-arc, Right-arc and Reduce respectively). We used Adam optimization technique with learning rate 0.01 and batch-size 256. In addition, the training iterates for 50 epochs. The experiment is conducted on two models, with and without adding attention layer. The result of the model to select the correct transition action for a given configuration is presented in figure 9. As shown in figure 9, the addition of attention layer helps the transition action prediction to achieve accuracy of 94.7. Whereas, the accuracy of transition action prediction with no attention layer is 91.5.

## 3.3 EVALUATION OF RELATIONSHIP TYPE PREDICTION

This experiment also used similar data to the pervious experiment converted into list of head-dependent pairs. When the dataset is converted in to head dependent pairs, we got 15,534 head-dependent pairs. Among this, 70 (with 10,874 head dependent pairs) are used for training the model and 30 (with 4,660 head-dependent pairs) are used for testing the model. We used the embedding layer and LSTM layers 256-output dimension. We used 36 relationship types including 'root' (to indicate the root word) and '-PAD-' (to indicate padded zero) for the output dimension. In this experiment, we use Adam optimization to train the network model with learning rate 0.01, 100 epochs and 256 batch-size. Out of 4660 head-dependent pairs, the model correctly predicts a dependency relation for 3,839 (81.4) head-dependent pairs. The result shows the accuracy of relationship type prediction if lower than that of transition action prediction. This is due to the limited size of Amharic treebank. The number of examples for relationship type prediction (with 36 classes) is less than the number of examples for transition action prediction (with four classes) in Amharic treebank.

## 3.4 ATTACHMENT SCORE

Unlabeled attachment score is the accuracy of the system to attach the correct headword with the dependent word without considering the relationship type. On the other hand, labeled attachment

| | Head-dependent pairs | Correctly attached | Scores in percent |
|---|---|---|---|
| Unlabeled attachment score (UAS) of the system | 4,660 | 4,266 pairs | 91.54 % |
| Labeled attachment score (LAS) of the system | 4,660 | 3,794 pairs | 81.4 % |

Figure 9: Attachment scores of the parser

score is the accuracy of the system to attach the correct headword to the correct dependent word and giving the correct relationship type. The parser is evaluated on 4,660 head dependent pairs, which is 30 of the total dataset. As depicted in figure 10 the parser scores 91.54 accuracy for unlabeled attachment and 81.4 accuracy for labeled attachment score.

## 4 DISCUSSION

The study attempts to design Dependency parser for Amharic language. For the purpose two models are designed, transition action prediction and relationship type prediction model. For transition action prediction, two models are evaluated. The first model is a network model with no addition of attention layer. This model correctly predicts 91.5 transition actions of the test configuration from Amharic treebank. On the other hand, the second model uses addition of attention layer. This system correctly predicts 94.7 of transition actions from the test configuration. From this experiment, we found that the use of attention layer on the network model increases the performance of the prediction model by 3.2. The performance of relationship type prediction was also evaluated by using test datasets from Amharic treebank and correctly predicts 82.4 of the dataset. From this experiment, we found that the accuracy of the relationship type prediction decreased relative to accuracy of transition action. This is because of the size of the dataset. Relationship type predictor have 36 classes to be predicted. This makes the number of examples for each classes in Amharic treebank to be smaller than the number of examples for transition action predictor, which have four classes. Due to this, the accuracy of transition action prediction becomes greater than accuracy of relationship type prediction. The system is evaluated using unlabeled attachment score (UAS) and labeled attachment score (LAS). From the experiment, the system correctly constructs 91.54 and 81.4 unlabeled and labeled attachments respectively. The challenge observed during in this study was the accuracy of the labeled attachment score. This is mainly due to two reasons. The first reason is the accuracy of the relationship type prediction is lower relative to accuracy of transition action prediction. This is further due to the size of Amharic treebank. The other reason is due architecture of the system. As described in section 6.3, the output of unlabeled dependency relation is passed to the relationship type (dependency relation) predictor model as an input. Because of this, the wrong unlabeled tree constructed in the previous step makes the predictor model to choose wrong dependency relation.

## 5 CONCLUSION

In this paper, an attempt is made to show effectiveness of transition based dependency parser for Amharic language. We constructed two network models for predicting arc-eager transition action and for predicting relationship types (dependency relations) using state of the art machine learning techniques. The use of the second model if to minimize the number of classes from 2n+2 to n, where n is the number of dependency relations in the language. This helps each dependency relation to have more examples from Amharic treebank and increase the accuracy of the prediction. We train the models 1574 sentences and scores 91.54 and 81.4 unlabeled and labeled attachment score respectively. Results show the parser can be made more effective using larger Amharic treebank. Event though, Amharic dependency parser reported in this paper has performed well, it is clear that there are pairs that are not correctly attached. Improving the size of Amharic treebank and addition of morphological features to the treebank would probably results in improvement on the performance of the proposed system, which is our next research direction

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
