# OpenReview forum: "Transition Based Dependency Parser for Amharic Language Using Deep Learning"
_ICLR.cc/2020/Conference — Reject_

### Official Review · AnonReviewer2 · 2019-10-22
**Official Blind Review #2**

**Rating:** 1

**Review:**

The paper claims to build a transition-based dependency parser for Amharic. To produce a parse tree, the paper takes a two-stage approach, first producing the structure of the tree, followed by annotating each individual arcs on the tree. The paper also explores attention and finds it useful for improving the performance.

I am giving a score 1, because the paper lacks precision and the contribution is not significant.

The major problem of the paper is precision. There is a lack of formal descriptions of the parser, e.g., the representation of the input and output, what each individual block in the figures are doing, how the arc-eager transition parser works. The text description is certainly helpful, but without formal descriptions, it is impossible to reproduce the results in the paper. Besides, without setting up a precise language/notation, the paper is not easy to follow.

In terms of contribution, the paper begins with the motivation that Amharic has certain language characteristics that are absent in languages, such as English, where most research effort has been spent. However, the paper still uses a parser for English, and does not utilize the special features of Amharic. Generic architectural modification, such as adding attention, has been explored, but this has little contribution on its own and has little to do with Amharic.

In addition to the above, other minor weaknesses of the paper includes typos, poor experimental practices, missing citations and related work, and other wording issues.

In particular, choosing the split of the data set that achieves the best result, as quoted below, is concerning.

"We experimented on 60/40, 70/30 and 80/20 train-test splitting ratios ... Therefore, we select 70/30 train-test splitting ratio throughout the experiments."

**Experience Assessment:**

I have read many papers in this area.

**Review Assessment: Checking Correctness Of Derivations And Theory:**

N/A

**Review Assessment: Checking Correctness Of Experiments:**

I carefully checked the experiments.

**Review Assessment: Thoroughness In Paper Reading:**

I read the paper thoroughly.

---

### Official Review · AnonReviewer1 · 2019-10-23
**Official Blind Review #1**

**Rating:** 1

**Review:**

* overview
- This paper describes a model for transition-based dependency parser, and tested on the Amharic treebank. It proposes a modification that is different from the common practice of combining the transition with dependency relation, namely to predict the transitions first, then the dependency relations.

* strengthens
 - Unfortunately, I can't find any strength in this paper. It is more like a course project than a research paper for ICLR.

* weaknesses
- The main idea of first unlabeled parsing then predicting labels is hardly novel, and it generally won't help, since the predicted labels could be useful features for determining the transitions. There is also no experiment to compare the proposed model to normal parsing model with 2n+2 predictions to support the author's claim.
- Generally, the architecture of the model is poorly explained. For example, it seems that the "embeddings" for the words are just one-hot index vectors (since the number of dimensions are the same as the vocabulary size), which is not what we mean by embeddings. It is also unclear what the dot product of the embeddings means. The attention layer is again not explained at all, e.g. what are the input and output.
- The splitting of train/test set does not make sense, the selection should not be influenced by the results at all. There is no reason not to use the standard split in UD.
- I don't see what is the challenge in the discussion section, since UAS is by definition lower than LAS.
- There numerous typos and grammar mistakes, and the citation format is broken.
- In many places, the quoted presumably Amharic words are not shown.

**Experience Assessment:**

I have published one or two papers in this area.

**Review Assessment: Checking Correctness Of Derivations And Theory:**

I assessed the sensibility of the derivations and theory.

**Review Assessment: Checking Correctness Of Experiments:**

I carefully checked the experiments.

**Review Assessment: Thoroughness In Paper Reading:**

I read the paper at least twice and used my best judgement in assessing the paper.

---

### Official Review · AnonReviewer4 · 2019-11-08
**Official Blind Review #4**

**Rating:** 1

**Review:**

The paper describes a dependency parser for Amharic text trained on the Yimam et al. 2018 treebank.

The proposed method is an unlabelled arc-eager transition-based dependency parser followed by a dependency label classifier. Both models are based on LSTM architectures. Unfortunately there is no clear description of these models, which makes it difficult to understand their structure.

The experimental section contains a questionable selection on the train/test split (which should have been done blindly before looking at the results) and lacks any baseline comparison with previous approaches. The authors argue that performing arc prediction and arc labeling in two separate stages is beneficial, but they don't compare with a method that perform both actions at the same time.

The motivation of the paper is also questionable: the introduction section describes some of the peculiarities of the Amharic language, such as being morphologically rich, but the proposed method seems completely generic and does not address or exploits any of these peculiarities. If the authors want to argue that Amharic would benefit from specialized parser architecture, the methods and results reported in this paper fail to provide evidence for this position.

The citation format is also broken.


**Experience Assessment:**

I have published one or two papers in this area.

**Review Assessment: Checking Correctness Of Derivations And Theory:**

N/A

**Review Assessment: Checking Correctness Of Experiments:**

I assessed the sensibility of the experiments.

**Review Assessment: Thoroughness In Paper Reading:**

N/A

---

### Decision · Program_Chairs · 2019-12-19

**Decision:**

Reject

**Comment:**

The paper builds a transition-based dependency parser for Amharic, first predicting transitions and then dependency labels. The model is poorly motivated, and poorly described. The experiments have serious problems with their train/test splits and lack of baseline. The reviewers all convincingly argue for reject. The authors have not responded.